# Comprehensive Analysis of N6-Methyladenosine Methylation in Transverse Aortic Constriction-Induced Cardiac Fibrosis Based on MeRIP-Seq Analysis

**DOI:** 10.3390/biomedicines13092092

**Published:** 2025-08-27

**Authors:** Shidong Liu, Pengying Zhao, Yuyuan He, Jieneng Wang, Bing Song, Cuntao Yu

**Affiliations:** 1The First Clinical Medical College, Lanzhou University, Lanzhou 730000, China; 120220903191@lzu.edu.cn (S.L.); 120220903481@lzu.edu.cn (P.Z.); Heyy18@lzu.edu.cn (Y.H.); 2Department of Cardiovascular Surgery, First Hospital of Lanzhou University, Lanzhou 730000, China; 120220902201@lzu.edu.cn; 3Department of Anesthesiology and Surgery, First Hospital of Lanzhou University, Lanzhou 730000, China; 4Department of Cardiovascular Surgery, Fuwai Hospital, National Center for Cardiovascular Diseases, Chinese Academy of Medical Sciences, Peking Union Medical College, Beijing 100037, China

**Keywords:** cardiac fibrosis, N6-methyladenosine methylation, MeRIP-seq, transverse aortic constriction

## Abstract

**Background:** The function and mechanism of N6-methyladenosine (m6A) methylation in pressure-overload cardiac fibrosis remains limited and unclear. This study aims to analyze and predict m6A modifications present in mouse hearts because of transverse aortic constriction (TAC). **Materials and Methods:** Twelve male C57BL/6 mice were randomly assigned to two groups, TAC group and sham group. The RNA Dot Blot assay was employed to evaluate the overall m6A methylation levels in both TAC and sham mice. The expression level of m6A-related enzymes were investigated through RT-PCR and Western blotting. MeRIP-seq and RNA-seq analyses were conducted to identify differentially modified m6A genes and mRNA expression genes. The protein–protein interaction (PPI) network was carried out to choose potential hub genes. Additionally, the transcription factor (TF)–microRNA (miRNA) coregulatory network and the drug–hub gene interaction network were built based on these hub genes. Furthermore, molecular docking simulations were also performed to analyze the interactions between drugs and hub genes. **Results:** Compared with the sham group, the TAC group demonstrated elevated levels of global m6A methylation. METTL3 and METTL14 were significantly upregulated, whereas FTO and ALKBH5 were significantly downregulated following TAC. MeRIP-seq analysis identified 17,806 m6A peaks associated with 9184 genes and 16,392 m6A peaks associated with 8550 genes in the TAC and sham groups, respectively. In conjunction with RNA-seq data, 66 genes were identified as exhibiting concurrent differences in both m6A methylation levels and mRNA expression. Six hub genes, *Cd33*, *Irf4*, *Nr4a2*, *Hspa1b*, *Nr4a1*, and *Adcy1*, were identified through the construction of a PPI network. The TF-miRNA coregulatory network contains six hub genes, 31 miRNAs, and 24 TFs. The drug–hub genes interaction network included five hub genes and 36 candidate drugs. **Conclusions:** The m6A modification is prevalent in TAC-induced cardiac fibrosis and significantly contributes to the fibrotic process by regulating critical genes. In the future, it may emerge as one of the potential cardiac fibrosis therapeutic targets.

## 1. Introduction

Cardiac fibrosis represents a common pathological alteration in various cardiovascular diseases at a specific stage [1,2], characterized by the pathological proliferation of cardiac fibroblasts (CFs), and the overexpression and unbalanced deposition of proteins related to collagen during the development of cardiac interstitial fibrosis [3]. CFs are activated into myofibroblasts in response to myocyte damage. These cells subsequently migrate to the injury site and synthesize collagen, including collagen I (COL-1), collagen III (COL-3), among other elements of the extracellular matrix (ECM) [4,5]. Cardiac fibrosis can be classified into two distinct types under varying pathological conditions. The first type, reparative fibrosis, is typically observed in cases of myocardial infarction, often associated with myocardial cell injury [6]. The second type, reactive fibrosis, occurs in patients experiencing cardiac pressure or volume overload (hypertension, aortic coarctation, or hypertrophic cardiomyopathy) and is marked by excessive proliferation of interstitial collagen without concurrent loss of cardiomyocytes [7,8]. The increase in the myofibroblast and increased collagen fiber deposition causes hypertrophic scars that eventually cause damaged cardiac recording syndrome and diastolic [9].

Epitranscriptomics has emerged as a critical and rapidly developing field in the study of cardiovascular diseases in recent years [10]. Among these modifications, N6-methyladenosine (m6A) methylation is the most prevalent type of post-transcriptional modification in all eukaryotic species [11,12], and it plays an important role in influencing various biological functions by modulating RNA stability, splicing, translation, and degradation [13]. The m6A modification is produced by a complex called “writers”, which includes Methyltransferase-like 3 (METTL3), Methyltransferase-like 14 (METTL14) [14,15], and Wilms tumor 1-associated protein (WTAP) [16]. This modification can be counteracted by demethylase enzymes, known as “erasers”, such as fat mass and obesity-associated protein (FTO) [17] and alkyl B homolog 5 (ALKBH5) [18]. The RNA-binding proteins, referred to as “readers”, including YTH domain-containing proteins 1/2 (YTHDC1/2) [19], YTH family proteins 1/2/3 (YTHDF1/2/3) [20], and Insulin-like growth factor 2 mRNA-binding proteins 1/2/3 (IGF2BP1/2/3) [21], selectively recognize m6A-modified RNAs in the cytoplasm and perform regulatory roles by modulating various aspects of RNA metabolism, including RNA stability, translation efficiency, transport mechanisms, and splicing processes. Emerging evidence has indicated that m6A modification is closely linked to cardiovascular diseases, particularly heart failure [22], myocardial infarction [23,24], and myocardial hypertrophy [25]. Nevertheless, the investigation into m6A modification in cardiac fibrosis induced by pressure overload remains limited, and its potential mechanism remains to be clarified.

In this study, we attempted to further investigate the association between m6A modification and pressure overload-induced cardiac fibrosis, and to elucidate the role of m6A modification in the progression of cardiac fibrosis in mice subjected to pressure overload. MeRIP-seq and RNA-seq technologies were employed to characterize the m6A-modified transcriptome profiles in heart tissues of sham and TAC mice, and to elucidate the biological functions of differentially m6A-modified and differentially expressed genes. Our study will contribute to the existing research about m6A modification in cardiac fibrosis and establish a solid foundation for further investigations into its therapeutic potential.

## 2. Material and Methods

### 2.1. Animals and Transverse Aortic Constriction (TAC) Model

Twelve male C57BL/6 mice (SPF grade, 7–8 weeks old) were procured from the Lanzhou Veterinary Research Institute of Chinese Academy of Agricultural Sciences (Lanzhou, China; license No. 2020-0002). The mice were subsequently randomly divided into either the TAC group or the sham group, with each group consisting of six animals. Prior to the surgical procedure, all mice were maintained in a restricted environment with a room temperature of 24 ± 1 °C, conditioned humidity of 55 ± 5%, and a 12/12-h light/dark cycle with free feeding and water for a minimum of seven days.

Anesthesia was administered by delivering a mixture of pure oxygen and volatile isoflurane using an anesthesia machine. Induction of anesthesia was achieved with an isoflurane concentration of 2–3% for 2–3 min, followed by maintenance of anesthesia at an isoflurane concentration of 1.5–2%. The mice in the TAC group were secured in a supine posture on the surgical platform, and a midline incision approximately 10 mm in length was performed extending from the mid-neck to the sternum to facilitate access to the mediastinum. Using microsurgical instruments, the pretracheal muscles, mediastinal fat, and thymus tissue were meticulously dissected and retracted to expose the aortic arch. Ligation of the aortic arch using 6-0 polypropylene sutures between the brachiocephalic trunk and the left common carotid artery through the placement of a 27-gauge blunt-end needle ensured that constriction resulted in a stenosis with an approximate diameter of 0.4 mm. Upon confirmation of the successful ligation, the needle was carefully withdrawn, and the mediastinum was meticulously closed in layers, starting with the muscular layer and followed by the skin. Surgery involving the mice in the sham group was similar, only that the aorta was not ligated. Specifically, only the aortic arch was exposed and visualized without any constriction. Subsequently, the mediastinum was closed in the same manner as in the TAC group. All mice were permitted to recover on a heated pad post-procedure. Following recovery, they were returned to the same environmental conditions as those prior to surgery, with unrestricted access to food and water.

Four weeks following TAC, mice in both the TAC and sham groups were euthanized under isoflurane anesthesia. The ARRIVE guidelines have been followed in all the procedures. Additionally, all experimental procedures were approved by the Animal Ethics Committee of the First Hospital of Lanzhou University, China, on 11 March 2025 (Approval Number: LDYYLL2025-122).

### 2.2. Echocardiographic Assessment

Four weeks after surgery, mice were anesthetized with an inhalational anesthetic, and their heart rates were continuously monitored and maintained within the range of 400 to 500 beats per minute. Echocardiographic assessments were performed by technicians who were blinded to the experimental groups, utilizing a high-frequency ultrasound system attached to the images of the small animal (Mylab X5 Vet, BersinBio, Guangzhou, China). Standard echocardiographic views, including the left ventricular long-axis, short-axis, and aortic arch views, were systematically acquired. Key measurements included the following: ejection fraction (EF), fractional shortening (FS), the interventricular septum diastolic thickness (IVS.d), interventricular septum systolic thickness (IVS.s), left ventricular internal diameter in diastole (LVID.d), left ventricular internal diameter in systole (LVID.s), left ventricular end-diastolic volume (LVEDV), left ventricular end-systolic volume (LVESV), and left ventricular posterior wall diastolic thickness (LVPWD). For the TAC and sham groups, statistical significance was evaluated using an unpaired Student’s *t*-test performed with the SPSS software (version 26.0; SPSS Inc., Chicago, IL, USA). A *p*-value < 0.05 was considered to indicate statistical significance (* *p* < 0.05, ** *p* < 0.01, *** *p* < 0.001).

### 2.3. Histology Analysis

The heart weight normalized to body weight (HW/BW), along with heart volume and outer diameter, was measured under standardized conditions following euthanasia of the mice. Masson’s trichrome staining was conducted on histological sections derived from paraffin-embedded cardiac tissues. The paraffin-embedded cardiac tissues were sectioned into slices with a thickness of 5 µm and subsequently underwent deparaffinization through three consecutive 10-min immersions in xylene. This process was followed by a series of graded ethanol dehydration steps, each lasting 5 min, and concluded with a final 2-min rinse using distilled water. The sections were subsequently incubated in potassium dichromate overnight and subjected to heat treatment at 63 °C for 1 h. Thereafter, the sections were stained with ponceau fuchsin solution for 10 min and gently rinsed with distilled water. Collagen fibers were subsequently treated with phosphomolybdic acid until the color faded, followed by aniline blue staining for approximately 2 min. Dehydration was performed using a graded alcohol series, and clearing was achieved with a transparency agent prior to mounting with neutral gum. Microscopic images were captured using a micrographic imaging system.

### 2.4. RNA Dot Blot

Heart tissue was homogenized using TRIpure Total RNA Extraction Reagent (ELK Biotechnology, Hubei, China; Catalog #EP013), followed by the extraction of total RNA in accordance with the standardized protocol. The RNA was diluted in RNase-free water, denatured at 95 °C for 5 min, and immediately chilled on ice. Subsequently, it was transferred onto a methanol-activated PVDF membrane (Millipore Corporation, Burlington, MA, USA, Catalog #IPVH00010) and crosslinked using UV light for 10 min. The membrane was washed with Wash Buffer for 5 min, followed by blocking with 5% non-fat milk in PBS-T for 1 h at room temperature. It was then incubated overnight at 4 °C with the primary antibody (anti-m6A; Proteintech, Wuhan, China; Catalog #68055-1-Ig). The following day, the membrane was washed three times with 10 mL of Wash Buffer for 5 min each at room temperature, followed by incubation with a diluted secondary antibody in 10 mL of Blocking Buffer for 1 h. Subsequently, the membrane was washed four times with 10 mL of Wash Buffer for 5 min each. The membrane was incubated with the ECL substrate (Aspen Technology, Burlington, MA, USA, Catalog #AS1059) and subsequently visualized through chemiluminescence imaging.

### 2.5. Real-Time Polymerase Chain Reaction (RT-PCR)

The total RNA was extracted using the Total RNA Kit I (Omega Bio-Tek, Norcross, GA, USA, Catalog #R6834-01), following the user’s instruction. The concentration and purity of the RNA samples were assessed using UV spectrophotometry, with the requirement that the RNA concentration must exceed 100 ng/μL. Subsequently, 1 microliter of total RNA (TAC: 134 ng/μL; Sham: 148 ng/μL) was reverse-transcribed into complementary DNA (cDNA) when further amplification using the Evo M-MLV RT Premix kit (Accurate Biology, Changsha, China, Catalog #AG11728). RT-PCR assays were performed in a 25-microliter reaction volume with the β-actin gene serving as an internal control. The cycle threshold (Ct) values obtained from the PCR instrument were utilized for the relative quantification of the initial templates. The primer sequences used are listed in Table 1.

### 2.6. Western Blot Analysis (WB)

Total protein was extracted from 0.2 g of cardiac tissue using 200 μL of RIPA lysis buffer (Servicebio, Wuhan, China, Catalog #G2002). Following extraction, a 10% SDS-PAGE gel was prepared, and 30 μg of total protein was subjected to electrophoretic separation. The separated proteins were then transferred onto a polyvinylidene difluoride (PVDF) membrane (Millipore Corporation, Burlington, MA, USA, Catalog #IPVH00010) and blocked with 5% non-fat milk powder (Biosharp, Beijing, China, Catalog #BS102-500g) at room temperature after a 2-h incubation. For antibody incubation, the primary antibody was diluted in the manufacturer process and subsequently applied to the PVDF membrane using a pre-prepared working solution. The membrane was incubated overnight at 4 °C on a shaker inside the refrigerator. After incubation with the primary antibody, the PVDF membrane was washed three times with Tris-buffered saline containing Tween 20 (TBST; Solaibao, Beijing, China; Catalog #T8220). A horseradish peroxidase (HRP)-conjugated goat anti-rabbit secondary antibody was applied to the membrane and incubated for 2 h at room temperature with gentle shaking. The membrane was then subjected to additional washes to remove unbound antibodies. Finally, the PVDF membrane was developed using enhanced chemiluminescence (ECL) reagent (Biosharp Co., Ltd., Guangzhou, China, Catalog #BL520B) for exposure analysis. The antibodies used are listed in Table 2.

### 2.7. Methylated RNA Immunoprecipitation Sequencing (MeRIP-Seq)

Four weeks after surgery, MeRIP-Seq were performed for three random samples from each group to identify genome-wide regions with m6A modifications in mouse cardiac tissue.

Total RNA was extracted using the TRIzol (Thermo Fisher Scientific, Waltham, MA, USA, Catalog #15596026), following the manufacturer’s instructions. To ensure the removal of potential DNA contamination, the RNA sample was subjected to DNase I (New England Biolabs, Ipswich, MA, USA, Catalog #M0303L). The quality and purity of the extracted RNA were assessed by determining the A260/A280 absorbance ratio using the NanodropTM OneC spectrophotometer (Thermo Fisher Scientific, Waltham, MA, USA). The integrity of the RNA was further validated using the LabChip GX Touch system (Revvity, Shanghai, China). Subsequently, the concentration of the qualified RNA was quantified using the Qubit 3.0 Fluorometer in conjunction with the QubitTM RNA Broad Range Assay Kit (Thermo Fisher Scientific, Waltham, MA, USA, Catalog #Q10210). Total RNA was then enriched for polyadenylated RNA (polyA+ RNA) following the manufacturer’s instructions provided with the VAHTS mRNA Capture Beads (Vazyme, Nanjing, China, Catalog #N401).

For m6A-methylated RNA immunoprecipitation (MeRIP) experiments, enriched polyA+ RNA were fragmented to an average length of approximately 100 nucleotides using a 20 mM ZnCl_2_ treatment at 95 °C for 5–10 min. A portion (10%) of the fragmented RNA was reserved as an “Input” control, while the remaining fraction underwent m6A-specific immunoprecipitation (IP). The fragmented polyA+ RNA was incubated with specific anti-m6A-polyclonal antibody (Synaptic Systems, Göttingen, Germany, Catalog #202003) and RNasin (Promega, Madison, WI, USA, Catalog #N2615) at a final concentration of 40 U/µL. The mixture was subsequently incubated at 4 °C for 2 h to promote efficient and specific binding of the antibody to the m6A-modified RNA. Immunoprecipitation of the RNA–antibody complexes was performed using protein G magnetic beads (Thermo Fisher Scientific, Waltham, MA, USA, Catalog #88848) at 4 °C for 1 h under rotational conditions to ensure efficient capture of the m6A-RNA complexes. Subsequently, the protein G magnetic beads were extensively washed to eliminate non-specifically bound RNA, and the immunoprecipitated RNA was extracted using TRIzol (Thermo Fisher Scientific, Waltham, MA, USA, Catalog #15596026).

Using “Input” and IP RNA as the input, RNA-sequencing library preparation was conducted using selected components of KC^TM^ Digital mRNA Library Prep Kit (SEQHEALTH, Wuhan, China), according to the instructions of the kit. The library preparation involved the enrichment of PCR products containing fragments ranging from 200 to 500 bp. Each of the libraries would then be quantified and sequenced on DNBSEQ-T7 platform (MGI Tech Co., Shenzhen, China) using the PE150 sequencing mode. The raw sequence data were submitted to the Genome Sequence Archive (Genomics, Proteomics & Bioinformatics 2021) in the National Genomics Data Center (Nucleic Acids Res 2024), China National Center for Bioinformation/Beijing Institute of Genomics, Chinese Academy of Sciences (GSA: CRA023973), which is accessible to the public at https://ngdc.cncb.ac.cn/gsa (accessed on 20 May 2025) [26].

### 2.8. Differential Expression Analysis and Functional Enrichment Analysis

Differentially methylated m6A peaks (Diffpeaks) were screened using the criteria of |log2 Fold Change (FC)| > 1 and *p*-value < 0.05. Similarly, differentially expressed genes (DEGs) were determined with the same thresholds of |log2 FC| > 1 and *p*-value < 0.05. Diffpeaks and DEGs were intersected, and genes that exhibited both differential m6A methylation modification and mRNA expression were ultimately identified. Visualized representations of volcano plots, heatmaps, and Venn diagrams for Diffpeaks and DEGs were constructed using the R software (version 4.5.0) with the corresponding R packages “ggplot2”, “ComplexHeatmap”, and “VennDiagram”, respectively.

The Diffpeaks and DEGs were subsequently subjected to Gene Ontology (GO) enrichment analysis and Kyoto Encyclopedia of Genes and Genomes (KEGG) pathway analysis using the R software (version 4.5.0) with the corresponding R package “clusterProfiler”. GO enrichment analysis provides a functional classification system for genes and proteins, encompassing biological process (BP), molecular function (MF), and cellular component (CC). In contrast, KEGG enrichment analysis offers insights into identified pathways and their associated interactions. Enriched GO terms and KEGG pathways were selected based on an adjusted *p*-value threshold of less than 0.05.

### 2.9. Construction of Protein–Protein Interaction and Identification of Hub Genes

The protein–protein interaction (PPI) network for both differentially methylated and expressed genes was constructed using the Search Tool for the Retrieval of Interacting Genes (STRING) (http://string-db.org/, accessed on 8 June 2025), with a confidence score threshold of ≥0.4. Visualization of the PPI network was subsequently performed using the Cytoscape software (version 3.10.3) (https://www.cytoscape.org/, accessed on 8 June 2025) [27]. The cytoHubba methods, including Degree, Edge Percolated Component (EPC), Maximum Clique Centrality (MCC), and Maximal Neighborhood Component (MNC), were employed to identify hub genes within the PPI network [28]. Finally, external data validation of hub genes was conducted using two GEO datasets, GSE18224 and GSE5500.

### 2.10. Establishment of Transcription Factors (TFs) and miRNA Coregulatory Network

The hub genes identified above were uploaded to NetworkAnalyst (version 3.0, https://www.networkanalyst.ca/, accessed on 8 June 2025) for the construction of TF–miRNA–gene networks [29]. The TF–gene interaction network was established based on ENCODE, while the miRNA–gene interaction network was constructed using miRTarBase v9.0.

### 2.11. Construction of Drug–Hub Genes Interaction Network and Molecular Docking Simulation

The potential drugs or compounds targeting the hub genes were identified through an analysis using Drug Gene Interaction Database (DGIdb) version 3.0.2 (https://www.dgidb.org, accessed on 9 June 2025) [30]. DGIdb is a website whose results are a compilation of multiple drug–gene interaction databases. The Cytoscape software was used subsequently to visualize the drug–hub genes interaction network. The protein structures of these drugs and hub genes were retrieved from PubChem (https://pubchem.ncbi.nlm.nih.gov/, accessed on 9 June 2025) and UniProt (https://www.uniprot.org/, accessed on 9 June 2025), respectively. Finally, molecular docking simulations were performed on the platform CB-Dock2 (https://cadd.labshare.cn/cb-dock2/php/index.php, accessed on 9 June 2025) to generate spatial 3D structures based on the Docking score.

## 3. Results

### 3.1. TAC Surgery Induced Ventricular Remodeling and Cardiac Fibrosis in Mice

M-mode echocardiograms were utilized to assess the aortic arch constriction velocity and pressure gradient, as well as the left ventricular structure and function status in both the TAC and sham groups (Figure 1A,B). Compared with the sham group, the flow velocity and pressure gradient at the aortic arch constriction site in the TAC group were markedly elevated (*p* < 0.001), thereby validating the successful establishment of the mouse transverse aortic constriction model (Figure 1C,D). A comparison with the sham group revealed that the TAC group exhibited significant increases in IVS.d (*p* < 0.001), IVS.s (*p* < 0.001), LVID.d (*p* < 0.05), LVID.s (*p* < 0.01), LVEDV (*p* < 0.05), LVESV (*p* < 0.01), and LVPWD (*p* < 0.01). Conversely, EF (*p* < 0.001) and FS (*p* < 0.01) significantly decreased (Figure 1E–M). Furthermore, a significant difference in HW/BW (*p* < 0.05) was observed between the two groups (Figure 1N). Masson staining revealed increased collagen deposition in the cardiac tissue of mice subjected to TAC induction (Figure 1O). These findings confirm the successful establishment of the mouse model of cardiac fibrosis following TAC.

### 3.2. Excessive m6A Methylation and Alteration of m6A-Associated Modifying Enzymes

The RNA m6A Dot Blot analysis was performed to evaluate the global m6A methylation levels in myocardial tissues from both the TAC and Sham groups four weeks after surgery (Figure 2A). The results indicate that the levels of m6A methylation were significantly higher in the TAC group compared to the sham group (*p* < 0.05; Figure 2B), suggesting that m6A methylation is dynamically elevated and may serve as a critical factor in the progression of cardiac fibrosis. To further investigate the mechanisms underlying the observed alterations in m6A methylation, the RT-PCR analysis was conducted to compare the relative mRNA expression levels of four key m6A methylation modifying enzymes—METTL3, METTL14, FTO, and ALKBH5—between the sham and TAC groups. The mRNA methyltransferases *METTL3* (*p* < 0.001) and *METTL14* (*p* < 0.001) were significantly upregulated in the TAC group compared to the sham group, while the mRNA demethylases *FTO* (*p* < 0.01) and *ALKBH5* (*p* < 0.001) were significantly downregulated in the TAC group (Figure 2C). Additionally, the protein expression levels of METTL3, METTL14, FTO, and ALKBH5 were evaluated using WB analysis. The results demonstrate that the protein expressions of methyltransferases METTL3 (*p* < 0.001) and METTL14 (*p* < 0.001) were markedly upregulated in the TAC group. Conversely, the protein expressions of demethylases FTO (*p* < 0.001) and ALKBH5 (*p* < 0.01) were significantly downregulated (Figure 2D,E). These experimental results further substantiate the hypothesis that m6A methylation is upregulated by these enzymes during cardiac fibrosis induced by TAC.

### 3.3. General and Topological Characteristics of m6A Methylation Modification in Cardiac Tissue-Based MeRIP-Seq Analysis

The MeRIP-seq analysis of mRNA derived from mouse cardiac tissue identified 17,806 m6A peaks associated with 9184 genes in the TAC group and 16,392 m6A peaks associated with 8550 genes in the sham group (Figure 3A,B). All m6A methylation modification-related genes were categorized based on the number of m6A peaks associated with each gene. Notably, the majority of m6A-methylated coding genes in the TAC and sham groups existed in only one or two m6A modification sites (Figure 3C). Further analysis of the m6A peak length distribution indicated that the peak length of m6A was primarily concentrated around 200 bp, with no statistically significant difference detected between the TAC and sham groups (Figure 3D). Additionally, the quantitative analysis of genes in each group clearly demonstrated that the m6A modification peaks in the TAC group were significantly greater than those in the sham group. These findings collectively provide robust evidence that the level of m6A modification is markedly increased following TAC.

To investigate the topological structure of m6A modifications, metagene profiles were employed to visualize the distribution across different regions of mRNA in the two groups (Figure 3E,F). Moreover, the m6A peaks were categorized into four distinct regions, the 3′-untranslated region (3′-UTR), coding sequence (CDS), internal exons (npExon), and the 5′-untranslated region (5′-UTR), and the distribution proportions of m6A peaks in various functional regions of the gene were quantitatively analyzed (Figure 3G,H). The results demonstrate that m6A sites were predominantly enriched in the CDS (59.64% and 59.03%, respectively) and 3′-UTR regions (33.16% and 32.97%, respectively) in both the TAC and sham groups. The m6A locus map demonstrated a consistent and uniform distribution of m6A peaks across all chromosomes in both groups (Figure 3I,J). The HOMER package (version v5.1) was utilized to systematically identify consensus motifs located in the region flanking the m6A peak, thereby enabling the identification of classical RRACH (R = A/G and H = A/C/U) motif structures (Figure 3K,L). The results demonstrate that the top-ranked m6A motifs were characterized by the sequence “AGACGUA” in both the TAC (*p* = 1 × 10^−7^) and sham (*p* = 1 × 10^−5^) groups.

### 3.4. Conjoint Analysis and Functional Enrichment Analysis of MeRIP-Seq and RNA-Seq

To investigate the transcriptome profiles in the TAC and sham groups, RNA-seq (MeRIP-seq “Input” library) was employed to quantify the levels of DEGs. In comparison with the sham group, a total of 580 genes were identified as differentially expressed in the TAC group (|log2 FC| > 1 and adjusted *p*-value < 0.05), comprising 156 upregulated genes and 424 downregulated genes (Figure 4A). A heatmap comparing the TAC and sham groups was also generated to visualize the DEGs (Figure 4C). Additionally, comparing with the sham group, a total of 1466 Diffpeaks were identified as differentially expressed in the TAC group (|log2 FC| > 1 and adjusted *p*-value < 0.05) based on MeRIP-seq analysis, including 717 hypermethylated sites and 749 hypomethylated sites (Figure 4B). A heatmap was also generated to compare the TAC and sham groups, thereby visualizing the differential expression of these Diffpeaks (Figure 4D).

GO and KEGG analyses were performed to elucidate the biological functions and potential regulatory mechanisms of the identified DEGs and Diffpeaks. With GO analysis, DEGs and Diffpeaks were categorized into three functional domains: Biological Process (BP), Cellular Component (CC), and Molecular Function (MF) (Figure 4E,F). Furthermore, DEGs were significantly enriched in six KEGG pathways, including Cytoskeleton in muscle cells, Cytokine-cytokine receptor interaction, Calcium signaling pathway, Apelin signaling pathway, Insulin signaling pathway, and AMPK signaling pathway (Figure 4G). Additionally, Diffpeaks were markedly enriched in six KEGG pathways, including Ras signaling pathway, Focal adhesion, Breast cancer, Lysine degradation, Polycomb repressive complex, and Regulation of lipolysis in adipocytes (Figure 4H).

Next, through the integrated analysis of RNA-seq and MeRIP-seq data, all genes were categorized into four distinct groups as follows (Figure 4I): 25 genes with both MeRIP and mRNA downregulation, 1 gene with both MeRIP and mRNA upregulation, 18 genes with MeRIP downregulation but mRNA upregulation, and 22 genes with MeRIP upregulation but mRNA downregulation (Figure 4J).

### 3.5. Construction of PPI Network and Identification of Hub Genes

The PPI network of all genes exhibiting differences in both m6A levels and mRNA expression was constructed using the STRING database to analyze gene interactions and visualized using the Cytoscape software (Figure 5A). Additionally, the interaction number of each gene is presented (Figure 5B). To identify the hub genes from the PPI network, the cytoHubba plugin of the Cytoscape software was utilized. Based on the MCC algorithm (Figure 5C), the top 10 genes were identified as potential hub genes, including Stearoyl-CoA Desaturase 1 (*Scd1*), Glucokinase (*Gck*), Fatty Acid Synthase (*Fasn*), CD33 Molecule (*Cd33*), Interferon Regulatory Factor 4 (*Irf4*), Nuclear Receptor Subfamily 4, Group A, Member 2 (*Nr4a2*), E2F Transcription Factor 1 (*E2f1*), Heat Shock Protein 1B (*Hspa1b*), Nuclear Receptor Subfamily 4, Group A, Member 1 (*Nr4a1*), and Adenylate Cyclase 1 (*Adcy1*). Using the MNC algorithm (Figure 5D), the top 10 genes were similarly identified as potential hub genes, including Potassium Inwardly Rectifying Channel, Subfamily J, Member 3 (*Kcnj3*), *Scd1*, *Gck*, *Fasn*, *Cd33*, *Irf4*, *Nr4a2*, *Hspa1b*, *Nr4a1*, and *Adcy1*. According to the Degree algorithms (Figure 5E), the top 10 genes were selected as potential hub genes, including T-box 5 (*Tbx5*), *E2f1*, *Gck*, *Kcnj3*, *Cd33*, *Irf4*, *Nr4a2*, *Hspa1b*, *Nr4a1*, and *Adcy1*. Lastly, based on the EPC algorithm (Figure 5F), the top 10 genes were identified as potential hub genes, including *Scd1*, *Kcnj3*, *Fasn*, *Cd33*, *Irf4*, *Nr4a2*, *E2f1*, *Hspa1b*, *Nr4a1*, and *Adcy1*. The intersection of the genes obtained through the above four algorithms was analyzed, and six hub genes were identified: *Cd33*, *Irf4*, *Nr4a2*, *Hspa1b*, *Nr4a1*, and *Adcy1* (Figure 5G).

### 3.6. Verification of the Six Hub Genes and Establishment of TF-miRNA Coregulatory Network

The TAC data (GSE18224 and GSE5500) were downloaded from the GEO database for the purpose of conducting external data validation of the six hub genes. Among the six hub genes, the expression levels of five hub genes (*Adcy1*, *Irf4*, *Nr4a2*, *Nr4a1*, and *Cd33*) were significantly downregulated, whereas one hub gene (Hspa1b) was significantly upregulated in GSE5500 relative to the control (Figure 6A). This evidence is in accordance with our results. In the GSE18224 dataset, the expression levels of three hub genes (*Adcy1*, *Irf4*, and *Nr4a2*) were significantly downregulated, one hub gene (*Hspa1b*) was significantly upregulated, while the expression levels of two genes (*Nr4a1* and *Cd33*) exhibited no significant difference compared with the control (Figure 6B). These results align closely with our reported data.

TFs and miRNAs represent two distinct categories of gene expression regulators. While miRNAs control post-transcriptional gene expression, TFs regulate transcription by binding to promoter regions. The TF-miRNA co-regulation network was constructed using the Network Analyst tool. This regulatory network consisted of six hub genes, 31 miRNAs, and 24 TFs, comprising 81 edges and 61 nodes in total. Specifically, the degrees of *Nr4a2*, *Nr4a1*, *Hspa1b*, *Irf4*, *Adcy1*, and *Cd33* were 4, 25, 20, 10, 14, and 8, respectively (Figure 6C).

### 3.7. Drug–Hub Genes Interaction Network and Molecular Docking Simulation

Potential drugs or compounds targeting the six hub genes were identified via an analysis using the DGIdb database, and the drug–hub gene interaction network was visualized with the Cytoscape software. Nevertheless, no potential drugs or compounds specifically targeting the Irf4 gene have been identified to date (Figure 7A). The protein structures of these drugs and hub genes were retrieved from PubChem and UniProt, respectively. However, no related 3D structures were identified for potential drugs and compounds targeting the *Hspa1b* and *CD33* genes (Table 3). The molecular docking simulations were conducted using the CB-Dock2 platform, and spatial 3D structures were generated based on the docking scores (Figure 7B–N).

## 4. Discussion

In the present study, TAC mice exhibited ventricular remodeling, impaired cardiac function, and cardiac fibrosis. The over-modification of m6A occurrence in the TAC myocardial tissue was observed during the m6A Dot Blot experiment. Furthermore, RT-PCR and WB analyses demonstrated that the markedly upregulated of methyltransferases METTL3 and METTL14 and markedly downregulated of demethylases FTO and ALKBH5 at the mRNA and protein expression levels. Subsequently, the general and topological characteristics of m6A methylation modification were identified through MeRIP-seq analysis, revealing 17,806 m6A peaks associated with 9184 genes in the TAC group and 16,392 m6A peaks associated with 8550 genes in the Sham group. These findings were further integrated with RNA-seq data for comprehensive analysis. Ultimately, 66 genes that exhibited concurrent differences in both m6A methylation levels and mRNA expressions were selected for further investigation. Six hub genes, *Cd33*, *Irf4*, *Nr4a2*, *Hspa1b*, *Nr4a1*, and *Adcy1*, were identified through the construction of a PPI network based on these 66 genes, and TF-miRNA coregulatory network and drug–hub genes interaction network were established for these six hub genes. Additionally, molecular docking simulations were performed to analyze the interactions between the hub genes and candidate drugs.

This study demonstrated that cardiac fibrosis developed in mice four weeks following TAC surgery. Additionally, the total RNA showed over-modification of m6A in fibrotic heart, potentially due to the upregulated methyltransferases (METTL3 and METTL14) and downregulated demethylases (FTO and ALKBH5). These findings agree with what is recorded in the other literature. METTL3 knockdown suppressed CF proliferation, fibroblast–myofibroblast transition (FMT), collagen production, and the m6A-level modifications of fibrosis-related genes induced by TGF-β1 [23]. METTL3 downregulates the expression of AR via m6A-YTHDF2 pathway, thereby promoting glycolysis and proliferation of cardiac fibroblast, resulting in cardiac fibrosis [33]. In addition, METTL3 knockdown has also been found to reduce IGFBP3 expression and inhibit the activation of cardiac fibroblasts (CFs) and exacerbate cardiac fibrosis [34]. METTL14 knockdown reversed lipopolysaccharide-induced myocardial cell damage by modulating the stability of TRPM7 mRNA, thereby offering a promising novel therapeutic target for the treatment of septic cardiomyopathy [35]. On the other hand, an overexpression of FTO in heart failure mice shows specificity in the demethylation of cardiac transcripts to increase positive results in heart cardiac contractile integrity by reducing deterioration of cardiac contractility, reductions in fibrosis, and augmentation in angiogenesis [36]. Potentially, through inhibition of cardiac fibrosis and cardiomyocyte hypertrophy, overexpressions of FTO could lead to cardiac functional benefits in diabetic cardiomyopathy mice [37]. ALKBH5 is implicated in macrophage-to-myofibroblast transition (MMT), and the accompanying cardiac fibrosis and dysfunction caused by hypertension [38]. Therefore, it can be concluded that m6A modification is of importance in various types of cardiac fibrosis.

The abundant m6A modification sites were identified in TAC mouse cardiac tissues using MeRIP-seq technology. Analysis of mRNA m6A distribution characteristics revealed that most m6A-methylated coding genes contained only one or two m6A modification sites, with the length of these modification sites predominantly being approximately 200 bp, which is consistent with findings reported in previous studies [39]. The analysis of the topological structure of mRNA m6A modifications showed that m6A sites are predominantly located in the CDS and 3′-UTR [40]. As discussed in the literature review, methylation within the CDS can influence alternative splicing, while translational regulation may be affected by methylation near the stop codon [41]. The RRACH motif structures (R = A/G and H = A/C/U) have been demonstrated to be over-represented motifs of m6A methylation, which are highly conserved in humans [42]. Correspondingly, both the TAC and sham groups identified the top-ranked m6A motif “AGACGUA” as being identical in the present study, thereby confirming the successful identification of specific m6A sites. Based on a comprehensive conjoint analysis of MeRIP-seq and RNA-seq datasets, 66 genes exhibiting concurrent differences in both m6A methylation levels and mRNA expression profiles were identified for further investigation.

In this study, a PPI network was constructed, and six hub genes were identified using four distinct algorithms, including *Cd33*, *Irf4*, *Nr4a2*, *Hspa1b*, *Nr4a1*, and *Adcy1*. *Cd33*, a member of the Siglec (sialic acid-binding Ig-like lectin) family, can suppress the inflammatory response of macrophages as an inhibitory receptor [43]. Bidirectional Mendelian randomization analysis revealed an inverse association between the *Cd33*^+^ monocyte subset and the risk of idiopathic pulmonary fibrosis [44]. *Irf4* is an immune system-associated transcription factor predominantly expressed in immune cells [45]. Previous studies have reported that Irf4 restricts chronic kidney disease progression and kidney fibrosis following ischemia reperfusion injury, potentially by enabling M2 macrophage polarization and suppressing a Th1 cytokine response [46]. *Nr4a2*, an orphan nuclear receptor Nr4a family member, protects cardiomyocytes from myocardial infarction through inducing autophagy, and its overexpression restricts myocardial cells apoptosis by directly targeting p53 [47]. Additionally, *Nr4a2* has been reported to regulate cardiac fibrosis by inducing autophagic flux and inhibiting apoptosis in myocardial injury associated with metabolic syndrome (MetS) [48]. *Hspa1b*, a critical component of the heat shock protein 70 family, functions as a molecular chaperone protein that is abundantly expressed by cells under stress conditions such as heat, hypoxia, inflammation, and oxidative stress. Hspa1b knockdown increased the phosphorylation of Erk and induced cardiomyocyte hypertrophy, which could be effectively inhibited by an Erk-signaling pathway inhibitor [49]. *Nr4a1* belongs to the nuclear receptor subfamily and plays a crucial role in regulating glucose and lipid metabolism, inflammatory responses, and vascular homeostasis [50]. It has been reported that cytosporone B activates *Nr4a1*, which exerts an inhibitory effect on diabetic cardiac fibrosis [51]. Furthermore, preventing the cytoplasmic localization of *Nr4a1* can inhibit the TGF-β1/Smads signaling pathway, consequently reducing fibroblast proliferation and collagen production in isoprenaline (ISO)-induced cardiac fibrosis [52]. The role of *Adcy1* in cardiac fibrosis has been investigated in a mice model of HFpEF. Research findings indicate that heart tissue from HFpEF mice exhibits a pronounced fibrotic phenotype. The downregulation of *Adcy1* expression is believed to contribute to the pathological process of fibrosis by modulating fibrosis-associated pathways, such as the adenylyl cyclase pathway, thereby emerging as one of the potential therapeutic targets for HFpEF [53].

### Limitations

Our study investigates the relationship between m6A modification and pressure overload-induced cardiac fibrosis, further elucidating the role of m6A modification in the progression of cardiac fibrosis in mice subjected to pressure overload. However, several limitations must be acknowledged. First, the research was performed using a murine model, which might restrict the direct applicability of the findings to human cardiac fibrosis. Secondly, while this study integrated MeRIP-seq and RNA-seq data to predict downstream target genes of m6A modification, experimental validation of these targets was not performed. Third, although we have demonstrated that m6A modification is prevalent in TAC-induced cardiac fibrosis and that m6A-modifying enzymes play a crucial role in this process, the precise mechanism by which m6A modification regulates gene expression remains unexplored. Future studies should focus on validating these results in human samples to confirm clinical relevance and investigate the potential therapeutic implications of m6A methylation in cardiac fibrosis.

## 5. Conclusions

In summary, our study demonstrated that m6A methylation is extensively present and dynamically regulated in TAC-induced myocardial fibrosis based MeRIP-seq analysis, which contributes to the deterioration of cardiac function and left ventricular remodeling. Six hub genes, *Cd33*, *Irf4*, *Nr4a2*, *Hspa1b*, *Nr4a1*, and *Adcy1*, were identified through the construction of a PPI network. Additionally, TF-miRNA coregulatory network and drug–hub genes interaction network were established for these six hub genes. These findings provide a solid foundation for further exploration of the mechanisms through which m6A methylation regulates cardiac fibrosis.

## Figures and Tables

**Figure 1 biomedicines-13-02092-f001:**
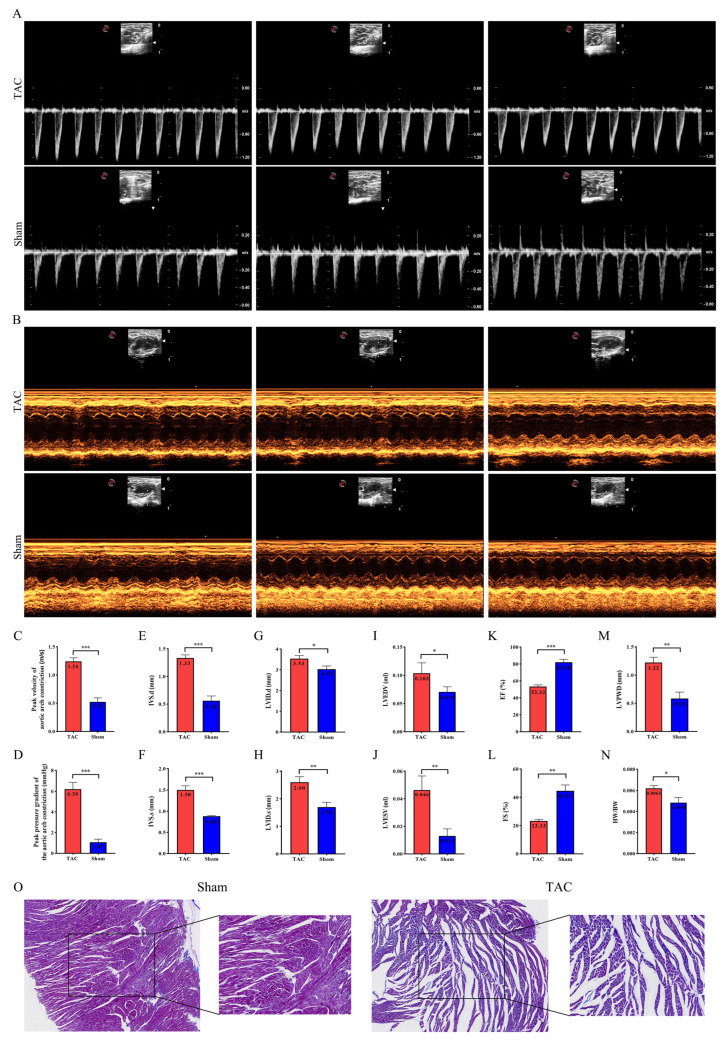
Ventricular remodeling and reduced cardiac function in mice after TAC. (**A**,**B**) Representative ultrasound images of mouse hearts in both groups. (**C**,**D**) Measurements of the flow velocity and pressure gradient at the aortic arch constriction in both groups, respectively (*n* = 6 each). (**E**–**N**) Measurements of IVS.d, IVS.s, LVID.d, LVID.s, LVEDV, LVESV, EF, FS, LVPWD, and HW/BW (*n* = 6 each). (**O**) Representative Masson images of mouse heart tissues to evaluate fibrosis; magnification: 100× and 200×. Data are presented as means ± SD. *, ** and *** represent *p* < 0.05, *p* < 0.01 and *p* < 0.001, respectively.

**Figure 2 biomedicines-13-02092-f002:**
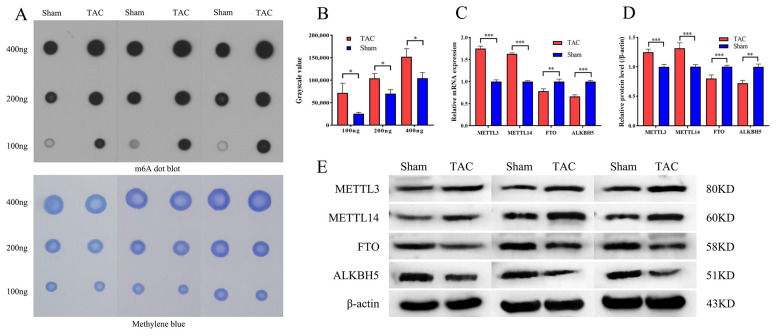
Excessive m6A methylation and alteration of m6A-associated modifying enzymes. (**A**) Representative image of RNA m6A Dot Blot in both groups. (**B**) The analysis of RNA m6A Dot Blot in both groups (*n* = 3 each). (**C**) The mRNA expression levels of *METTL3*, *METTL14*, *FTO*, and *ALKBH5* (*n* = 6 each). (**D**) The protein expression levels of METTL3, METTL14, FTO, and ALKBH5 (*n* = 6 each). (**E**) Representative WB image of METTL3, METTL14, FTO, and ALKBH5. Data are presented as means ± SD. *, ** and *** represent *p* < 0.05, *p* < 0.01 and *p* < 0.001, respectively.

**Figure 3 biomedicines-13-02092-f003:**
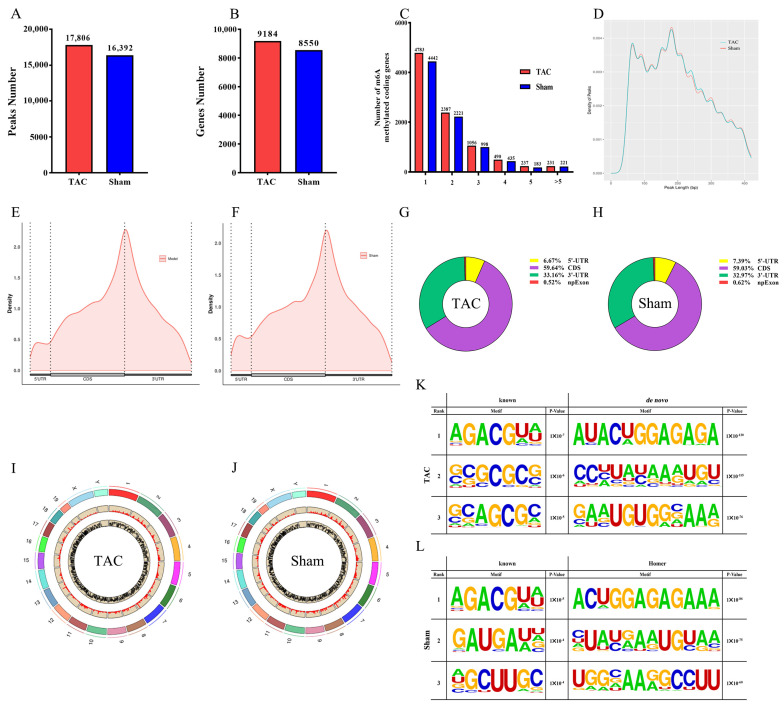
General and topological characteristics of m6A methylation modification. (**A**) The number of m6A peaks in both groups. (**B**) The number of m6A-modified genes in both groups. (**C**) The number of m6A peaks per gene in both groups. (**D**) The length of m6A peaks in both groups. (**E**,**F**) The density of m6A peaks in the 5′-UTR, CDS, and 3′-UTR regions of genes. (**G**,**H**) The distribution proportions of m6A peaks across each functional region of genes. (**I**,**J**) The distribution of m6A peaks in chromosome (the outermost circle represents the length abbreviation of the chromosome, the middle circle illustrates the distribution map of peak counts across each chromosome, and the innermost circle depicts the distribution map of peak enrichment folds for each chromosome). (**K**,**L**) The top three known and de novo Homer motifs in both groups with corresponding *p*-values.

**Figure 4 biomedicines-13-02092-f004:**
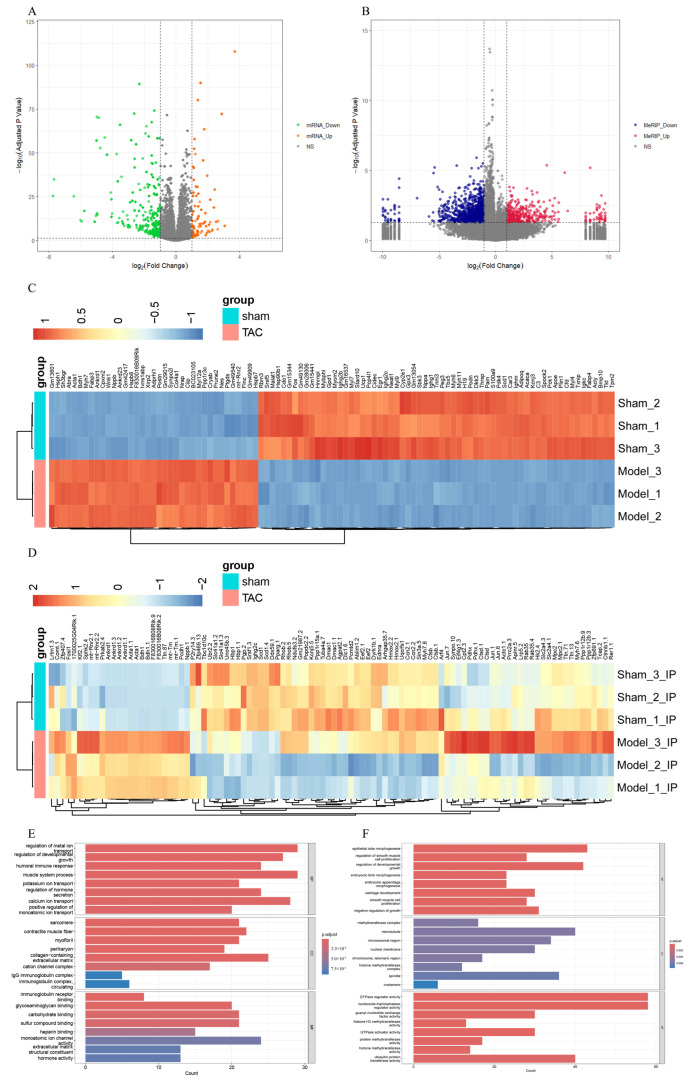
Conjoint analysis and functional enrichment analysis of MeRIP-seq and RNA-seq. (**A**) The volcano plot of DEGs. (**B**) The volcano plot of Diffpeaks. (**C**) The heatmap of DEGs. (**D**) The heatmap of Diffpeaks. (**E**) The bar chart of GO analysis in DEGs. (**F**) The bar chart of GO analysis in Diffpeaks. (**G**) The bar chart of KEGG analysis in DEGs. (**H**) The bar chart of KEGG analysis in Diffpeaks. (**I**) Four-quadrant graph exhibiting the distribution of genes with a significant alterations in both m6A and mRNA levels between the TAC and sham groups. (**J**) The Venn diagram of genes exhibiting both differential m6A and mRNA expression.

**Figure 5 biomedicines-13-02092-f005:**
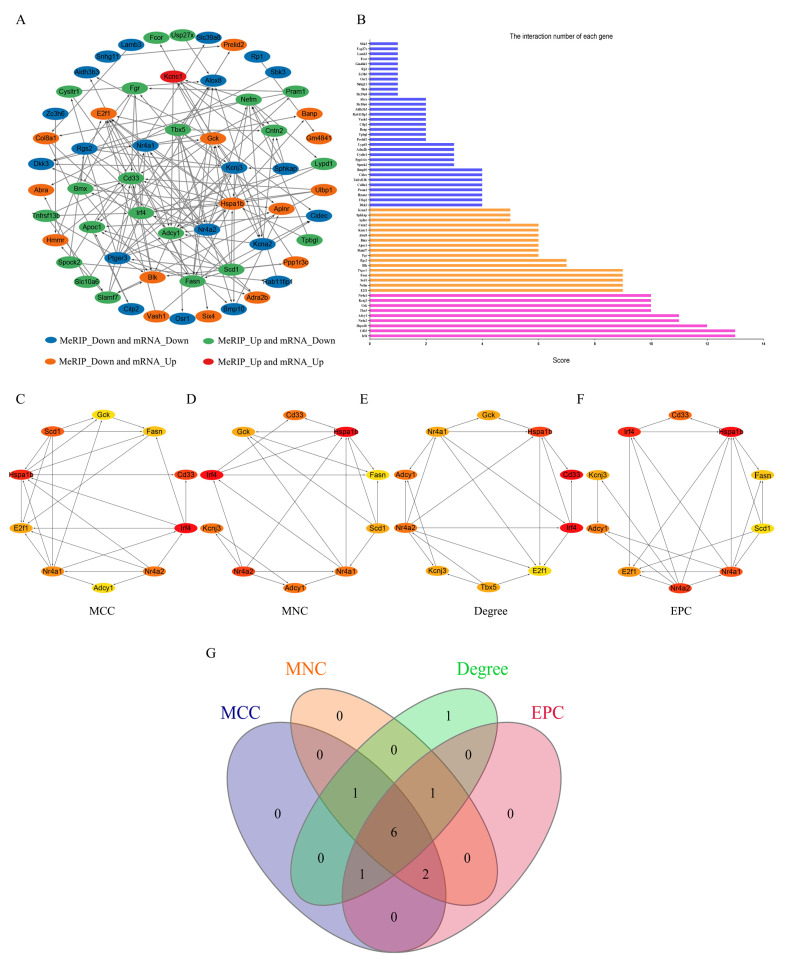
PPI network and hub genes. (**A**) The PPI network of all genes exhibiting differences in both m6A levels and mRNA expression. (**B**) The interaction number of each gene. (**C**) The top 10 hub genes of PPI network through the MCC algorithm. (**D**) The top 10 hub genes of PPI network through the MNC algorithm. (**E**) The top 10 hub genes of PPI network through the Degree algorithm. (**F**) The top 10 hub genes of PPI network through the MPC algorithm. (**G**) The Venn diagram of genes obtained through the four algorithms.

**Figure 6 biomedicines-13-02092-f006:**
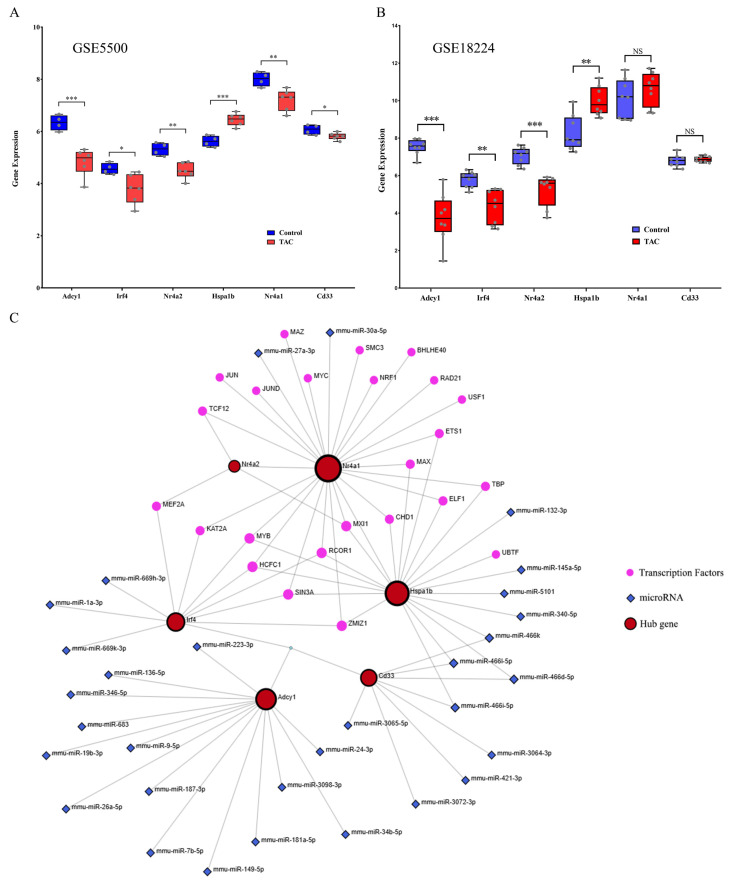
Verification of the hub genes and TF-miRNA coregulatory network. (**A**) The mRNA expression levels of the 6 hub genes were determined from the GSE5500 dataset. (**B**) The mRNA expression levels of the 6 hub genes were determined from the GSE18224 dataset. (**C**) TF-miRNA coregulatory network. NS, *, ** and *** represent no significant**,** *p* < 0.05, *p* < 0.01 and *p* < 0.001, respectively.

**Figure 7 biomedicines-13-02092-f007:**
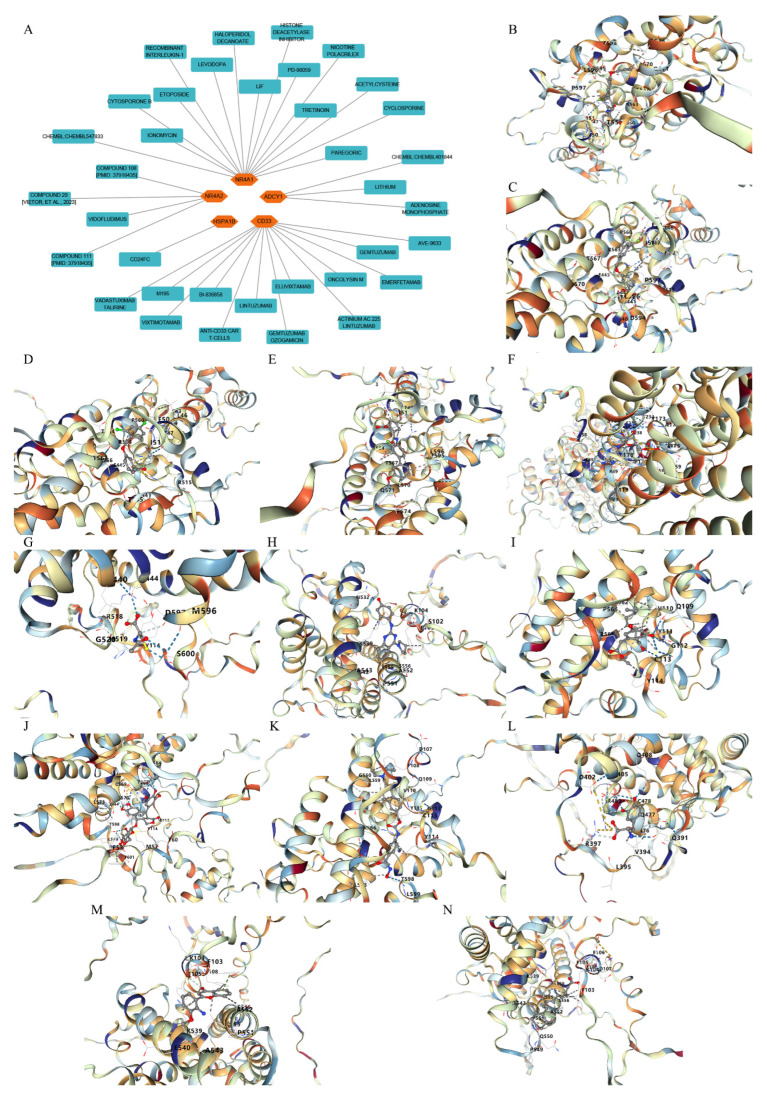
The drug–hub genes interaction network and molecular docking simulation. (**A**) The interaction network of drug–hub genes. (**B**–**E**) The molecular simulation docking of gene *Nr4a2* and drug Compound **29** (Vietor et al., 2023 [31]) (Docking score = −8.3 kcal/mol), Compound **108** PMID 37918435 [32]) (Docking score = −7.3 kcal/mol), Compound **111** (PMID 37918435 [32]) (Docking score = −6.8 kcal/mol), and Vidofludimus (Docking score = −7.8 kcal/mol), respectively. (**F**) The molecular simulation docking of gene Adcy1 and drug Adenosine Monophosphate, Docking score = −9.1 kcal/mol. (**G**–**N**) The molecular simulation docking of gene *Nr4a1* and drug Acetylcysteine (Docking score = −4.1 kcal/mol), CHEMBL547833 (Docking score = −6.9 kcal/mol), Cytosporone B (Docking score = −5.9 kcal/mol), Etoposide (Docking score = −8.4 kcal/mol), histone deacetylase inhibitor (Docking score = −7.5 kcal/mol), Levodopa (Docking score = −6.2 kcal/mol), PD-98059 (Docking score = −7.3 kcal/mol), and Tretinoin (Docking score = −6.7 kcal/mol), respectively.

**Table 1 biomedicines-13-02092-t001:** Primer sequence of real-time PCR.

Genes	Primer Sequence 5′-3′	Amplified Fragments/bp
*β-actin*	CATCCGTAAAGACCTCTATGCC AACATGGAGCCACCGATCCACA	171
*METTL3*	GGACTCTGGGCACTTGGATTTA	248
	CAGGTGCATCTGGCGTAGAG	
*METTL14*	GACTGGCATCACTGCGAATG	126
	AGGTCCAATCCTTCCCCAGA	
*ALKBH5*	CACGTTGACCCCATCCACAT	240
	CCTGAGAATGATGACCGCCC	
*FTO*	TCTGTGTGTTGGGTGTCCTTT	143
	AAAACGACAGCGGTGCTTAC	

**Table 2 biomedicines-13-02092-t002:** The antibodies used in Western blot analysis.

Antibody	Dilution	Source	Article No.
METTL14	1/1000	Fine Test, Wuhan Fine Biotech Co., Wuhan, China	FNab10824
METTL3	1/1000	Fine Test, Wuhan Fine Biotech Co., Wuhan, China	FNab05139
ALKBH5	1/1000	Fine Test, Wuhan Fine Biotech Co., Wuhan, China	FNab00314
FTO	1/1000	Fine Test, Wuhan Fine Biotech Co., Wuhan, China	FNab09787
β-actin	1/5000	ImmunoWay, Plano, TX, USA	YT0099

**Table 3 biomedicines-13-02092-t003:** The molecular formula and structure of potential drugs and compounds.

Gene	Drug Name	Molecular Formula	Structure	Docking Score (kcal/mol)
*Nr4a2*	Compound **29** Vietor et al., 2023 [31]	C_22_H_18_FNO_4_	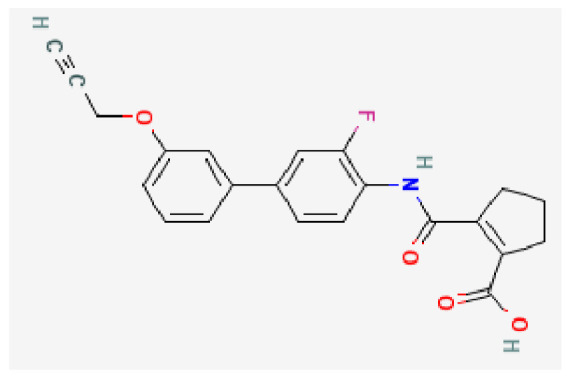	−8.3
	Compound **108** PMID 37918435 [32]	C_15_H_11_F_3_O_3_	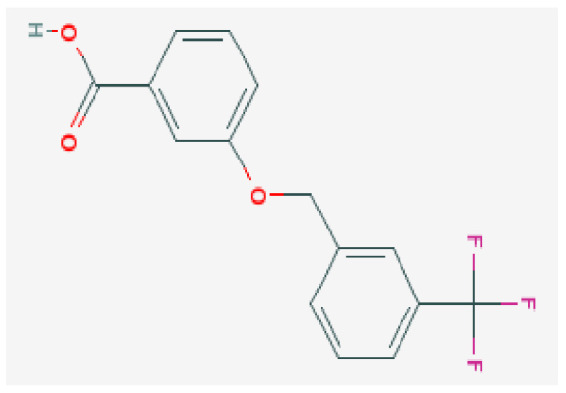	−7.3
	Compound **111** PMID 37918435 [32]	C_14_H_10_Cl_2_O_3_	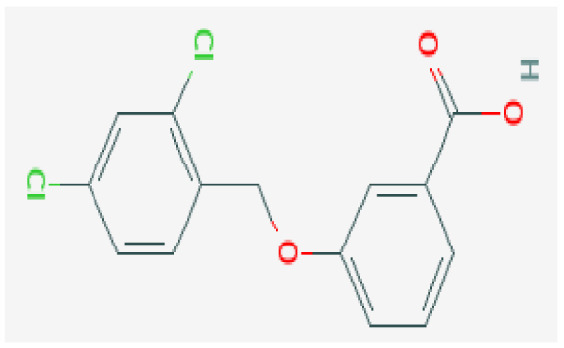	−6.8
	Vidofludimus	C_20_H_18_FNO_4_	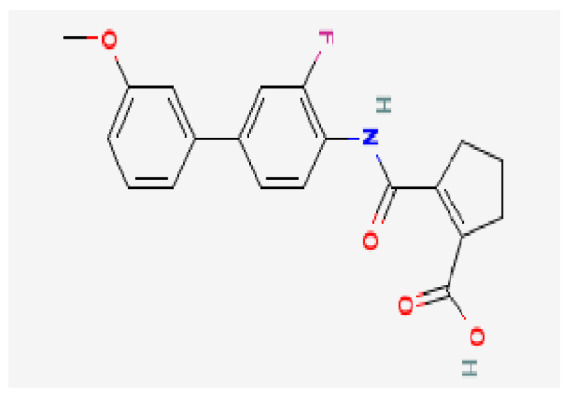	−7.8
*Adcy1*	Adenosine Monophosphate	C_10_H_14_N_5_O_7_P	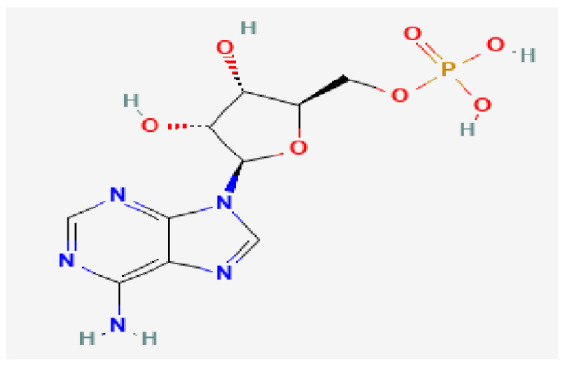	−9.1
*Nr4a1*	Acetylcysteine	C_5_H_9_NO_3_S	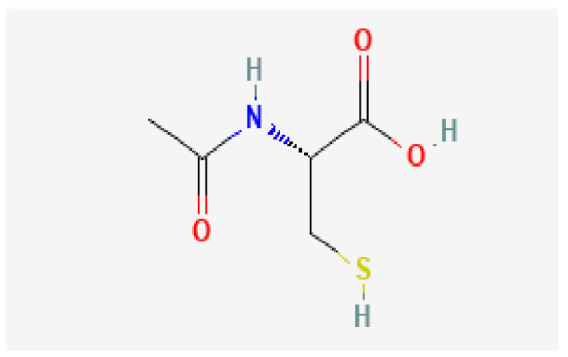	−4.1
	CHEMBL547833	C_18_H_19_ClN_4_O	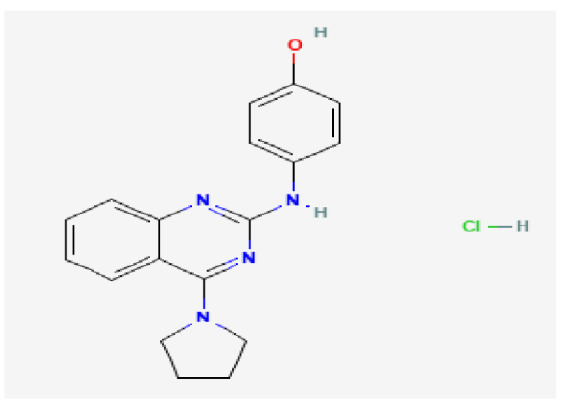	−6.9
	Cytosporone B	C_18_H_26_O_5_	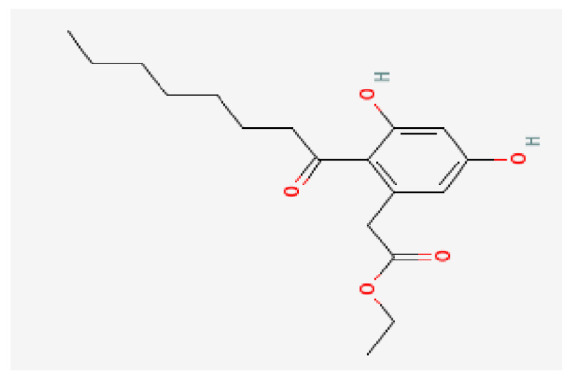	−5.9
	Etoposide	C_29_H_32_O_13_	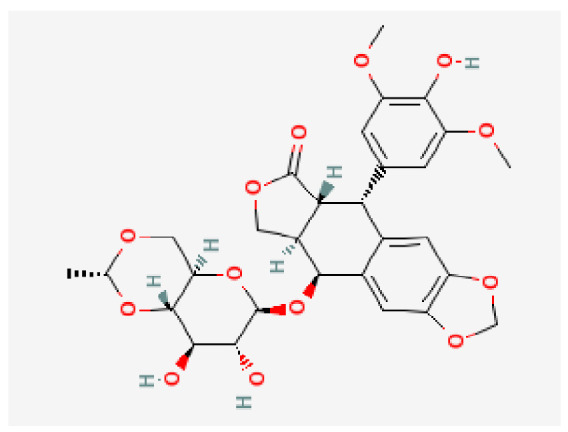	−8.4
	Histone deacetylase inhibitor	C_24_H_28_ClN_3_O_4_	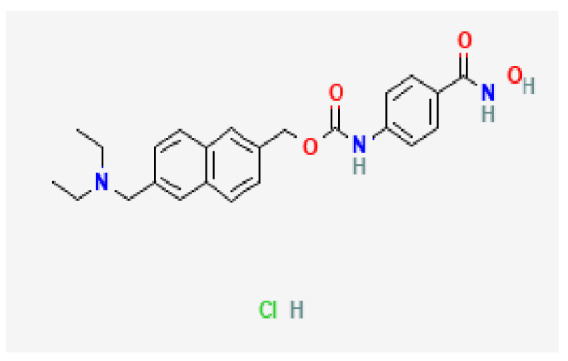	−7.5
	Levodopa	C_9_H_11_NO_4_	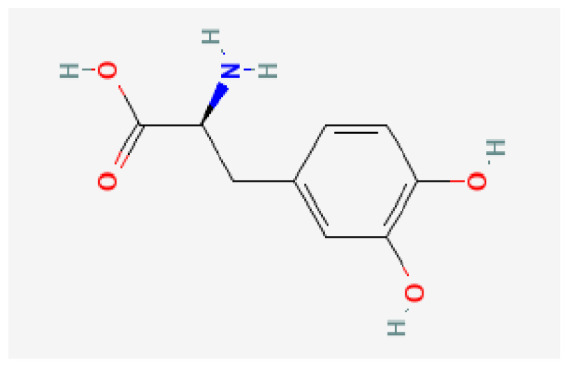	−6.2
	PD-98059	C_16_H_13_NO_3_	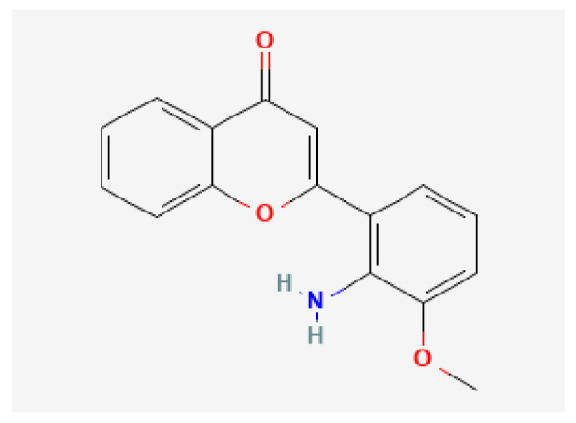	−7.3
	Tretinoin	C_20_H_28_O_2_	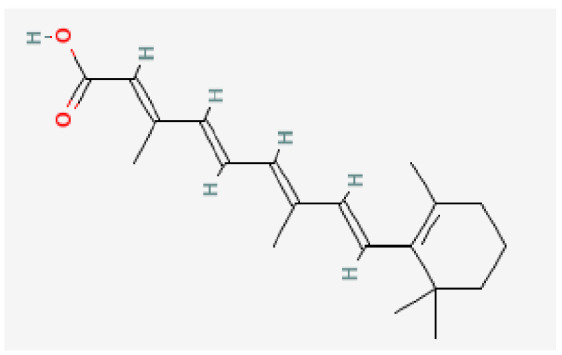	−6.7

## Data Availability

The raw sequence data were deposited in the Genome Sequence Archive (Genomics, Proteomics & Bioinformatics 2021) in the National Genomics Data Center (Nucleic Acids Research 2024), China National Center for Bioinformation/Beijing Institute of Genomics, Chinese Academy of Sciences (GSA: CRA023973), and are publicly accessible at https://ngdc.cncb.ac.cn/gsa (accessed on 20 May 2025).

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
