# Peer review of "Comprehensive Analysis of N6-Methyladenosine Methylation in Transverse Aortic Constriction-Induced Cardiac Fibrosis Based on MeRIP-Seq Analysis"

_biomedicines, 2025, doi:10.3390/biomedicines13092092_

Round 1
Reviewer 1 Report
Comments and Suggestions for Authors
- Abstract
In the methods section, there was no mentioning of mice surgery and groupings.
- Methods
“one microliter of total RNA was reverse-transcribed into complementary DNA”. Better to indicate RNA concentration instead of volume.
Line 206: I didn’t understand the rationale behind subjecting extracted RNA to different assays: Nanodrop, LabChip GX Touch system, Qubit 3.0 Flu- 213, Fluorometer, and QubitTM RNA Broad Range Assay Kit. Wasn’t nanodrop sufficient?
Author Response
Dear Reviewer,
We sincerely thank you for your thorough and constructive comments on our manuscript. We have carefully addressed all the points raised and made corresponding revisions to the manuscript. Below, we provide a point-by-point response to each comment.
Comments 1:
Abstract
In the methods section, there was no mentioning of mice surgery and groupings.
Response 1:
Thank you for pointing this out. We agree with this comment. Therefore, mice surgery and groupings have been included in the Abstract section (page 2 and line 27-28).
Comments 2:
Methods
“one microliter of total RNA was reverse-transcribed into complementary DNA”. Better to indicate RNA concentration instead of volume.
Response 2:
Agree. The RNA concentration has been included in the Methods section (page 10 and line188-189).
Comments 3:
Line 206: I didn’t understand the rationale behind subjecting extracted RNA to different assays: Nanodrop, LabChip GX Touch system, Qubit 3.0 Flu- 213, Fluorometer, and QubitTM RNA Broad Range Assay Kit. Wasn’t nanodrop sufficient?
Response 3:
Thank you for pointing this out. We agree with this comment and make the following statement.
In MeRIP-seq, the Nanodrop TM OneC spectrophotometer was used to measure the A260/A280 ratio to evaluate the purity of the extracted RNA.
In MeRIP-seq, it is essential to use the LabChip GX Touch system to assess RNA integrity, as the m⁶A antibody specifically recognizes intact RNA fragments. Degraded RNA may lose its modified sites, which can result in false-negative outcomes or biased enrichment.
- MeRIP-seq generally involves randomly fragmenting RNA to an average length of approximately 100 nucleotides, followed by immunoprecipitation of the m⁶A-modified fragments. If the RNA has already been degraded into short fragments smaller than 50 nucleotides, not only will the 18S and 28S ribosomal RNA bands disappear, but the m⁶A sites may also be disrupted, which can prevent antibody binding and lead to a loss of signal.
- RNA degradation can generate many "unnatural" RNA ends containing 5′-PO₄ and 3′-OH groups. These ends may exhibit non-specific binding to magnetic beads or antibodies, potentially leading to the appearance of false positive peaks.
- Both excessively small and excessively large fragments may compromise the efficiency of junction connection, thereby reducing library complexity and increasing the prevalence of repetitive sequences, ultimately undermining the accuracy of m⁶A site quantification.
The LabChip GX Touch ensures that only structurally intact RNA, with preserved modification sites, proceeds to downstream immunoprecipitation and sequencing through high-sensitivity, objective quantification and rapid-throughput microfluidic electrophoresis, thereby ensuring the accuracy and reproducibility of m⁶A detection.
In MeRIP-seq, it is essential to use the Qubit 3.0 Fluorometer and the QubitTM RNA Broad Range Assay Kit to reconfirm the concentration of qualified RNA. This step is critical because the m⁶A antibody enrichment process is highly sensitive to the actual quantity of intact RNA molecules that can participate in the m⁶A antibody reaction system, referred to as the "true input amount." Among available quantification methods, the Qubit system is uniquely capable of delivering accurate, absolute quantification at the nanogram level under conditions of high purity and low background interference. The Qubit 3.0 Fluorometer and QubitTM RNA Broad Range Assay Kit play a critical role in MeRIP-seq experiments by ensuring that the initial RNA quantity falls within a strictly defined range, which is essential for reliable m⁶A immunoprecipitation and high-quality library construction, ultimately enhancing the accuracy and reproducibility of the resulting data.
We appreciate your overall positive evaluation and agree with the need for these improvements.
We believe these revisions have substantially improved the manuscript, and we are grateful for your guidance in refining the work.
We thank the reviewer again for the valuable feedback and look forward to your positive consideration of our revised manuscript.
Sincerely,
Cuntao Yu

Reviewer 2 Report
Comments and Suggestions for Authors
The submitted article for review "Comprehensive Analysis of N6-Methyladenosine Methylation in Transverse Aortic Constriction-Induced Cardiac Fibrosis Based on MeRIP-seq Analysis" is undoubtedly relevant. However, during the review process, several remarks arose:
-
In the Introduction or Discussion section, provide a more detailed explanation of how m6A methylation is linked to fibrosis development. Are there any supporting data from other researchers?
-
In Figure 1, more representative images should be presented, or adjustments should be made to the magnification.
-
In Figures 4 and 5, the text is completely illegible
Author Response
Dear Reviewer,
We sincerely thank you for your thorough and constructive comments on our manuscript. We have carefully addressed all the points raised and made corresponding revisions to the manuscript. Below, we provide a point-by-point response to each comment.
Comments 1:
In the Introduction or Discussion section, provide a more detailed explanation of how m6A methylation is linked to fibrosis development. Are there any supporting data from other researchers?
Response 1:
Thank you for this important observation. We agree with this comment. Therefore, we have provided a more detailed explanation and supporting data in the Introduction and Discussion section
Comments 2:
In Figure 1, more representative images should be presented, or adjustments should be made to the magnification.
Response 2:
Agree. Thank you for pointing this out. We have presented more representative images in Figure 1 and adjusted the magnification of the representative images.
Comments 3:
In Figures 4 and 5, the text is completely illegible.
Response 3:
We are very sorry for the inconvenience caused to you. We have enhanced the resolution of Figures 4 and 5 in the revised manuscript to ensure better clarity when viewed or printed. Besides, all the original pictures of the images in the manuscript have been uploaded as attachments to the manuscript.
We appreciate your overall positive evaluation and agree with the need for these improvements.
We believe these revisions have substantially improved the manuscript, and we are grateful for your guidance in refining the work.
We thank the reviewer again for the valuable feedback and look forward to your positive consideration of our revised manuscript.
Sincerely,
Cuntao Yu
